# Detection of Chronic Blast-Related Mild Traumatic Brain Injury with Diffusion Tensor Imaging and Support Vector Machines

**DOI:** 10.3390/diagnostics12040987

**Published:** 2022-04-14

**Authors:** Deborah L. Harrington, Po-Ya Hsu, Rebecca J. Theilmann, Annemarie Angeles-Quinto, Ashley Robb-Swan, Sharon Nichols, Tao Song, Lu Le, Carl Rimmele, Scott Matthews, Kate A. Yurgil, Angela Drake, Zhengwei Ji, Jian Guo, Chung-Kuan Cheng, Roland R. Lee, Dewleen G. Baker, Mingxiong Huang

**Affiliations:** 1Department of Radiology, University of California at San Diego, San Diego, CA 92121, USA; dharrington@health.ucsd.edu (D.L.H.); rtheilmann@health.ucsd.edu (R.J.T.); adangeles@health.ucsd.edu (A.A.-Q.); arobb@health.ucsd.edu (A.R.-S.); tao.song@megin.fi (T.S.); z2ji@health.ucsd.edu (Z.J.); rrlee@health.ucsd.edu (R.R.L.); 2Research, Radiology, and Psychiatry Services, VA San Diego Healthcare System, San Diego, CA 92161, USA; 3Department of Computer Science and Engineering, University of California, San Diego, CA 92093, USA; p8hsu@eng.ucsd.edu (P.-Y.H.); ckcheng@ucsd.edu (C.-K.C.); 4Department of Neurosciences, University of California, San Diego, CA 92093, USA; slnichols@health.ucsd.edu; 5ASPIRE Center, VASDHS Residential Rehabilitation Treatment Program, San Diego, CA 92110, USA; lu.le@va.gov (L.L.); carl.rimmele@va.gov (C.R.); scmatthews@health.ucsd.edu (S.M.); 6Department of Psychological Sciences, Loyola University, New Orleans, LA 70118, USA; kyurgil@loyno.edu; 7VA Center of Excellence for Stress and Mental Health, San Diego, CA 92161, USA; dgbaker@health.ucsd.edu; 8Department of Psychiatry and Behavioral Medicine, University of California, Davis, CA 95817, USA; aidrake@ucdavis.edu; 9School of Computer Science, Nanjing University of Posts & Telecommunications, Nanjing 210023, China; guoj@njupt.edu.cn; 10Department of Psychiatry, University of California, San Diego, CA 92093, USA

**Keywords:** diffusion tensor imaging, mild traumatic brain injury, chronic traumatic encephalopathy, machine learning, support vector machines

## Abstract

Blast-related mild traumatic brain injury (bmTBI) often leads to long-term sequalae, but diagnostic approaches are lacking due to insufficient knowledge about the predominant pathophysiology. This study aimed to build a diagnostic model for future verification by applying machine-learning based support vector machine (SVM) modeling to diffusion tensor imaging (DTI) datasets to elucidate white-matter features that distinguish bmTBI from healthy controls (HC). Twenty subacute/chronic bmTBI and 19 HC combat-deployed personnel underwent DTI. Clinically relevant features for modeling were selected using tract-based analyses that identified group differences throughout white-matter tracts in five DTI metrics to elucidate the pathogenesis of injury. These features were then analyzed using SVM modeling with cross validation. Tract-based analyses revealed abnormally decreased radial diffusivity (RD), increased fractional anisotropy (FA) and axial/radial diffusivity ratio (AD/RD) in the bmTBI group, mostly in anterior tracts (29 features). SVM models showed that FA of the anterior/superior corona radiata and AD/RD of the corpus callosum and anterior limbs of the internal capsule (5 features) best distinguished bmTBI from HCs with 89% accuracy. This is the first application of SVM to identify prominent features of bmTBI solely based on DTI metrics in well-defined tracts, which if successfully validated could promote targeted treatment interventions.

## 1. Introduction

Traumatic brain injury (TBI) can lead to long-term physical, cognitive, and behavioral dysfunction [1,2,3]. Improvised explosive devices are a common cause of TBI in military personnel [4,5,6]. Among TBIs in the United States military, 82% are mild TBI (mTBI), with blast exposure being the leading cause [7]. The pathophysiology of blast-related mTBI (bmTBI) is not thoroughly understood, and controversy remains over its long-term consequences [8,9]. Currently, there are no optimal rehabilitation interventions, in part, because the mechanism (s) of injury are not fully understood [10]. Mild TBIs typically go undetected by standard diagnostic approaches, even in individuals with persistent post-concussive symptoms (PCS) [11,12,13,14]. This deficiency underscores the need for more sensitive techniques that characterize the predominant pathophysiology, which is essential for accurate diagnosis and targeted interventions.

In recent years, diffusion tensor imaging (DTI) has been used to investigate mTBI [15,16,17,18], as the pathophysiology disrupts microstructural white matter connections [19,20,21], which are not visible using conventional anatomical MRI (i.e., T1-weighted, T2-weighted, T2*-weighted, and fluid-attenuated inversion recovery (FLAIR) sequences) [22,23]. DTI measures in vivo water diffusion in tissues that are more restricted perpendicular as opposed to parallel to the fiber directions within axon bundles, thus permitting the local microstructural characteristics of tissue to be inferred [24,25]. Common metrics derived from anisotropic diffusivity are fractional anisotropy (FA), mean diffusivity (MD), axial diffusivity (AD), and radial diffusivity (RD) [15,26,27]. Although some research reports significant differences in patterns of diffusion metrics between healthy control (HC) and bmTBI cohorts, findings across studies have been inconsistent [15,24,28,29], partly owing to differences in the post-injury timeframe, which have included acute (<1 month post-injury) [30,31], subacute/chronic (1 month to 1 year) [32,33], and remote (>1 year) injuries [34,35,36]. Studies of military bmTBI, also differ in terms of whether control participants are civilians or military and whether they have a history of blast exposure [37]. Also, many studies include bmTBI participants with comorbidities that can influence DTI findings, including depression and post-traumatic stress disorder (PTSD) [34,36,37,38]. DTI analytic approaches also differ amongst studies [34,39,40], including the use of metric(s) that can elucidate the pathogenesis of white-matter injury, which requires analyses that evaluate not only the degree of anisotropy, but also the components (AD and RD eigenvalues) of FA and MD [41,42]. Collectively, these factors can influence DTI outcomes.

Despite the ability of DTI to detect white-matter abnormalities in mTBI at the group level, its low sensitivity (18–41%) limits its use for individual diagnosis. Most studies of mTBI have employed a single DTI metric (e.g., FA) in specific white-matter regions, which can be reliable for comparing two groups, but not for diagnostic categorization [15,24,28]. For example, FA sensitivities for identifying individuals with mTBI were 41% for the anterior corona radiata, 29% for the uncinate fasciculus, 21% for the inferior longitudinal fasciculus, 18% for the cingulum bundle, and 29% for the genu of corpus callosum [43]. Another study found that only 26% of mTBI patients could be accurately identified based on FA in the cingulum, uncinate fasciculus, and anterior limb of the internal capsule [44]. The lack of diagnostic sensitivity impels the need for leveraging more powerful approaches. Specifically, analytic approaches that evaluate multiple DTI metrics from white-matter tracts throughout the brain are be needed to understand optimal combinations of metrics from different brain regions that better distinguish mTBI patients from controls [15,45,46]. In this regard, machine learning (ML) approaches may pave the way for developing personalized applications of neuroimaging data [25]. One such approach is support vector machines (SVM), which can handle classification challenges associated with multivariate datasets where the number of dimensions can exceed the number of cases [47], which is common in medicine where obtaining large datasets for some clinical populations can be challenging.

The present study leveraged SVM modeling to uncover optimal combinations of DTI metrics from multiple tracts that best distinguished white-matter abnormalities in subacute/chronic bmTBI participants from healthy controls (HC). As our study aimed to build a diagnostic model for future verification, feature selection and model evaluation were performed using the same sample, as in many studies of mTBI [29,39,48,49,50]. Although the use of nonindependent samples can introduce bias into model evaluation, approaches to limiting bias included the selection of the best features to guard against overfitting of the data [51]. Rather than focusing on a single metric, clinically relevant features for modeling were selected using voxelwise tract-based analyses that identified group differences throughout white-matter tracts in five DTI metrics, which together elucidate the pathogenesis of white-matter injuries. Subsequently, SVM models with cross-validation were applied to the metrics of these thresholded clusters to identify sets of consensus features that best classified each participant as either bmTBI or HC. We hypothesized that the SVM method would enhance sensitivity beyond that of any individual metric from the tract-based analysis, thereby distinguishing prominent features of mild blast-related injuries that could potentially provide a promising diagnostic model to guide future validation studies.

## 2. Methods

### 2.1. Participants and Assessments

Participants were 19 HC without a history of TBI and 20 bmTBI with persistent post-concussive symptoms (PCS). All participants were male combat-deployed active-duty service members or veterans from the Operation Enduring Freedom/Operation Iraqi Freedom era. The study was approved by institutional review boards of the VA San Diego Healthcare System, San Diego, CA, USA and Naval Health Research Center, San Diego, CA, USA. Participants gave written informed consent for all procedures. Characteristics of the two groups are described in Table 1. The number of days post-injury ranged between 130 and 758. Fifteen bmTBI participants were tested 4 months to 1-year post-injury (subacute/early chronic) and 5 participants were tested 1 to 2 years post-injury (remote) 28. MRI scans were reviewed by a board-certified neuroradiologist (RRL) and were negative in all bmTBI participants (see Appendix A for examples). Two bmTBI participants had a PTSD diagnosis. There were no group differences in age or years of education.

The diagnosis of bmTBI was based on standard Veterans Affairs/Department of Defense diagnostic criteria [52]: (1) loss of consciousness <30 min or transient confusion, disorientation, or impaired consciousness immediately after the trauma; (2) post-traumatic amnesia <24 h; and (3) an initial Glasgow Coma Scale [53] between 13–15. Using a modified Head Injury Symptom Checklist (HISC) [54], ongoing PCS were assessed for 21 categories (Table 2). Participants with persistent symptoms in at least three of the categories were recruited into the study. Exclusion criteria for study participation included: (1) prior head injury; (2) history of other neurological, developmental or psychiatric disorders (e.g., brain tumor, stroke, epilepsy, bipolar disorder, major depressive disorder prior to injury, or self-reported diagnosis of learning disability); (3) substance or alcohol abuse (DSM-V criteria) within the six months prior to the study, based on a clinical interview; (4) currently taking sedative neuroleptics and hypnotic medications; and (5) suicidal thoughts or wishes, confirmed in follow-up risk assessment.

Subtests from the Delis-Kaplan Executive Function System (D-KEFS) [55] and the Wechsler Adult Intelligence Scale Version 3 (WAIS-III) [56] respectively assessed executive function and processing speed (Table 1), two cognitive domains that are vulnerable in mTBI [57]. The Number-Letter Sequencing and Category Switching subtests test to measure cognitive flexibility and set shifting. The Letter Fluency and Category Fluency subtests assess phonemic and semantic retrieval. Processing speed was assessed by the Symbol Search and Digit Symbol Coding subtests. Tests were performed in a single session, within one week of the MRI session. All scores were age-corrected scaled scores from normative data.

### 2.2. Imaging Protocols

Imaging was conducted at the UCSD Radiology Imaging Laboratory using a General Electric (Chicago, IL, USA) 1.5 T HD Excite Twin Speed MRI system with an eight-channel head coil. T1-weighted high-resolution anatomic images were collected (Spoiled Gradient Recalled, TR = 10.6 milliseconds (ms), TE = 43.8 ms, FOV = 24 cm, slice thickness = 1.2 millimeter (mm), NEX = 1, Flip Angle = 10°). Two-dimension T2-weighted images were AC/PC aligned and acquired with a fractional fast spin echo sequence (FRFSE-XL, TR = 6462, TE = 86 ms, number of slices = 33 with no gap, slice thickness = 4 mm, FOV = 25.6 cm, matrix = 320 × 256). The imaging protocol also included axial T2*-weighted images (TR = 500 ms, TE = 30 ms, number of slices = 33, slice thickness/space = 4 mm/4.4 mm, matrix = 320 × 192) and axial FLAIR images (TR = 8800 ms, TI = 2200 ms, TE = 80 ms, FOV = 25 cm, number of slices = 25 with no gap, slice thickness = 4 mm, matrix = 320 × 224). DTI images were acquired in the axial plane using a single-shot EPI sequence with diffusion encoding along 51 directions, b-value = 1000 s/mm^2^, five non-diffusion weighted image (b0), 56 slices, thickness = 2.5 mm, TR = 15.1 s, TE = 80.4 ms, matrix = 96 × 96 mm (automatically re-gridded onto a 128 × 128 matrix), FOV = 24 cm, and voxel size 1.875 × 1.875 × 2.5 mm^3^. The same number of slices with no gap were used for all participants. Study participants underwent two DTI acquisitions to increase the signal-to-noise ratio and ensure at least one artifact-free scan. The acquisition time for each DTI scan was 13 min.

### 2.3. DTI Analyses

DTI analyses were conducted using the Functional Magnetic Resonance Imaging of the Brain Software Library (FSL) version 4.1.5 (http://www.fmrib.ox.ac.uk/fsl, accessed on 10 February 2019) (Appendix A for details). For each participant, images from each DTI acquisition were concatenated into a single dataset and corrected for eddy currents and motion by registering all brain volumes to the first non-diffusion weighted image using a rigid body transformation (no rotation). By avoiding rotations in our motion correction (i.e., we only considered motion in the X, Y, and Z direction), issues with changing the encoding direction prior to the analysis were avoided. Each DTI scan was visually inspected to ensure the absence of susceptibility artifacts and significant rotation in the brain volume. If significant motion was present, the entire brain volume was removed from the dataset (Appendix A for details). Data were fit to the diffusion tensor model for each voxel using FMRIB’s Diffusion Toolbox (FDT) [58]. The tensor was then diagonalized to three eigenvalues which are used to calculate FA, RD, MD, and AD. We also analyzed the $\frac{\mathrm{AD}}{\mathrm{RD}}$ ratio to evaluate the shape (length over width) of the fiber [59], since FA alone is insufficient to confirm specific pathologic mechanisms of mTBI [42] such as inflammation or axonal cytotoxic edema in mTBI [50,60,61,62].

Voxel-wise statistical analyses were carried out using track-based statistical analyses (TBSS) [63]. First, a study-specific FA template image was produced by aligning all participants’ FA data to Montreal Neurological Institute (MNI) 152 standard space by non-linear registration. From the FA template, a white matter skeleton was created with an FA threshold > 0.20 to represent the core of white-matter tracts common to all participants. Group differences in FA within the skeleton were tested by applying the FSL general linear model (GLM) tool. Permutation testing (5000) was then conducted to establish voxel-by-voxel *p*-values to quantify group differences in the DTI parameter (FA). In each permutation, group differences were evaluated by nonparametric two-sample *t*-tests on MNI152 standard space. Threshold Free Cluster Enhancement [64] was used to identify clusters of FA that significantly differed between the two groups (familywise error rate, FWE, corrected *p* < 0.05) after permutation statistical testing). Using the same registration matrix for FA (standard TBSS procedure), AD, RD, MD, and AD/RD were registered to the FA skeleton in MNI152 standard space and then voxel-wise tests for group differences in these metrics were conducted.

### 2.4. SVM Analyses with Cross Validation

DTI features for the SVM analyses were selected using the following procedures (Appendix A). First, clusters that showed significant group differences in the most representative DTI metric were identified as voxels located within a white-matter tract defined by the International Consortium of Brain Mapping (ICBM) DTI-81 atlas [65]. We selected FA as the representative metric since the largest number of voxels were found for this metric in tests for group differences. Next, we included the remaining DTI metrics that showed significant group differences. The features for the SVM analyses for MD, AD, RD, and AD/RD were those lying within the representative FA cluster mask. The mean DTI metric (i.e., averaged across voxels within the cluster mask) was computed for each cluster that showed significant group differences, which served as the inputs into the SVM models.

Stepwise SVM-based models were implemented to distinguish bmTBI from HCs based on linear combinations of multiple diffusion metrics from clusters in multiple white-matter regions as identified above (2.4). Appendix A show a flowchart and visualization of the procedure to construct the classification model. The complete procedure includes one algorithm and three routines (see Appendix A). The core concept of the procedure is to try all possible combinations of the DTI metric set and select the optimal combination (s). For each subset of metrics, features for classification are selected by removing the least relevant feature recursively.

Cross-validation was used to evaluate the performance of a built model with a small dataset [66]. Cross-validation is a robust technique that provides enhanced confidence of ML classification results through the validation of testing datasets with the model built on training datasets [49,51]. Thus, combining SVM together with cross-validation can potentially generate a robust model of features that best separate bmTBI participants from HC. In this study, we randomly leave five subjects out (k = 5) in each validation (10,000) because normally, the segmentation of validation data in small datasets is around 10−15% [67]. The process randomly selects five samples out of the combined bmTBI and HC cohorts and calculates the averaged feature weights, correctness, sensitivity, and specificity across validations (Appendix A).

### 2.5. Correlations between SVM Model Features and Behavioral Variables

The Statistical Package for the Social Sciences (SPSS version 28) was used to perform Pearson correlations, which tested for relationships between the DTI features in the optimal SVM model and neuropsychological variables that showed significant group differences (false discovery rate adjusted). These correlation analyses were conducted for the combined bmTBI and HC groups. For the bmTBI group only, the same DTI features were also correlated with total number of PCS and days post-injury.

## 3. Results

### 3.1. Neuropsychological Test Performance

Mean neuropsychological test performances fell within the average range for both groups (Table 1). Scores on Category Switching (total correct responses) were significantly lower in the bmTBI than the HC group (*p* < 0.05). A subthreshold trend for group differences was observed for Letter Fluency (*p* = 0.055).

### 3.2. TBSS Results

Among the five metrics, we found significant group differences in FA, RD, and AD/RD (Table 3), but not MD or AD. FA (Figure 1a) and AD/RD (Figure 1c) were significantly higher in the bmTBI group than the HC group in the genu and body of the corpus callosum (gCC/bCC), the anterior corona radiata (ACR), the anterior and posterior limbs of the internal capsule (ALIC/PLIC), and the superior corona radiata (SCR) (*p* < 0.05, FWE corrected). RD was reduced in the bmTBI group in all aforementioned regions except the left PLIC (Figure 1b).

For all white-matter features listed in Table 3, power analyses were performed using G*Power [68]. All features showed power > 0.80 at two-tailed α = 0.05 level, with the exceptions of the left ACR RD feature (power = 0.71) and the left PLIC RD feature (power = 0.37).

### 3.3. SVM with Cross-Validation Results

Figure 2 (top) displays the tracts from which aberrant DTI features in bmTBI from the TBSS analyses (Table 3) were extracted for the SVM analyses. We chose FA as the representative metric since FA had the largest number of voxels showing significant group differences in TBBS analyses. We then selected the voxels lying within significantly increased FA and the ICBM-DTI-81 white-matter atlas as the mask for generating FA, RD, and AD/RD features for stepwise SVM modeling. Altogether, 29 features were selected for SVM modeling (Table 3).

Figure 2 (bottom) illustrates how SVM separates the two groups using the two dominant DTI features (i.e., largest weightings) to draw the separation. Appendix A demonstrates the averaged error in each step after recursively removing the least contributing feature. Experiments to build the SVM model were conducted with both the inclusion and exclusion of the AD/RD metric. Without AD/RD, in which the four standard DTI metrics were considered, the FA-only model was the best SVM model, showing 75% accuracy, 82% sensitivity, and 79% specificity. Despite group differences in RD, the RD-only model achieved 72% accuracy, 77% sensitivity, and 64% specificity. Furthermore, when both FA and RD were input features in an SVM model, the FA-only model was output by the SVM algorithm. With both FA and AD/RD as input features, performance of the SVM model was boosted, showing 89% accuracy, 90% sensitivity, and 88% specificity.

Table 4 lists the five DTI features and associated weightings that were selected by our best stepwise SVM model. The top three features based on their weightings were dominated by AD/RD for left ALIC, gCC, and right ALIC. The fourth and the fifth features were FA of the right ACR and the left SCR. In the SVM model, AD/RD for the left ALIC took on the heaviest weighting.

### 3.4. Correlations between the SVM Features and Behavioral Variables

In the entire sample, weighted combinations of DTI features from the best SVM model did not significantly correlate with neuropsychological test scores, nor did individual features from this model. SVM features were also not correlated with total number of PCS and days post-injury in the bmTBI group.

## 4. Discussion

We identified an optimal linear combination of DTI features that distinguished bmTBI patients from HC with 89% overall accuracy, 90% sensitivity, and 88% specificity in cross-validation tests. The most discerning DTI features of bmTBI consisted of abnormally increased FA in right ACR and left SCR together with elevated AD/RD in the gCC and ALIC bilaterally. Despite high classification accuracy, SVM features failed to correlate with neuropsychological test scores or clinical variables, possibly due to relatively intact executive functioning of the bmTBI group on most tests [69], although statistical tests may also have been underpowered given the relatively small sample sizes.

To our knowledge, this is the first study to use ML to distinguish bmTBI and HC solely based on DTI features. Importantly, feature selection was constrained by outcomes from tests of group differences in five DTI metrics to select features that were clinically relevant for diagnostic modeling and to avoid overfitting of the data [51]. Vergara’s group also used an SVM approach to characterize white-matter features in noncombat mTBI patients [48]. Both FA and resting-state functional network connectivity (rsFNC) served as features in their models. They reported that functional connectivity within the default mode network provided the best classification accuracy (84%) followed by FA (75.5%) as indexed by a Z score that represented FA abnormalities throughout spatially heterogeneous white-matter relative to the control group. Their findings are similar to ours in terms of both increased FA in mTBI and our 75% correct classification for FA alone, despite measuring FA from clusters within skeleton tracts. Yet modeling of both rsFNC and FA failed to boost classification accuracy (74.5%) in their study, whereas our classification accuracy (89%) was improved by combining the AD/RD ratio and FA features. The performance of our SVM model indicates that multivariate combinations of different diffusion metrics in well-defined tracts are potentially potent markers of bmTBI. Our model also identified the most discerning abnormal anatomical tracts that together generated the best SVM model, which was not investigated by Vergara’s group.

### 4.1. Abnormal Diffusivity in bmTBI

Increased FA coupled with decreased RD are probably due to the deformation of the microstructural white-matter regions caused by blast, which can stretch and twist the axons [70,71]. This distortion is likely to tighten white-matter fiber bundles and disrupt neurofilaments and microtubules that elongate down the axons. Secondary to the compacted tightened axons and disrupted neuronal substances, the axons swell and reduce extracellular space [72], thereby restricting water diffusion perpendicular to axons [72]. Such restriction limits the movement of water molecules, leading to abnormally decreased RD, increased AD/RD, and by inference, increased FA [50]. Our results align with studies of acute mTBI in adolescents [73,74], acute/subacute mTBI [50] and bmTBI with comorbid PTSD [30], and acute/chronic sports-related concussions [75]. Rats and swine with blast-induced TBI also show increased FA [70,76]. Moreover, rats with concussive injury show increased FA and decreased RD, but normal AD relative to rats with sham injury [42].

The post-injury timeframe in our bmTBI cohort was somewhat constrained (four months to two years) to reduce heterogeneity on a variable that influences neuroimaging outcomes. For example, abnormally increased and decreased FA have been found in bmTBI during acute post-injury periods of less than four months [28,30,31] and after one to 12 months post injury [30,33]. Yet for people with remote post-injury times longer than two years, FA is typically decreased in bmTBI [34,36,40,77,78,79]. Collectively, FA appears to decrease for post-injury periods longer than two years, whereas for shorter post-injury durations, discrepant findings could be explained by different fiber recovery stages in subacute injuries [71] and the long term misalignments of fibers and axonal degeneration in early-chronic mTBI [80]. Future research is needed to systematically track longitudinal changes in white-matter diffusivity including DTI components (AD, RD, AD/RD) that can elucidate the recovery from pathological mechanisms of blast injuries.

### 4.2. Optimal Features of bmTBI

Of the 29 DTI features selected for SVM modeling, five features in tracts that best distinguish bmTBI from HCs involved the ALIC, ACR, SCR, and gCC, which aligns with the preponderance of abnormal diffusivity in anterior regions of in TBI [81,82,83,84]. Indeed, frontal areas are the most vulnerable to blast exposures owing to the design of military helmets [81,83]. Corpus callosum injury is also common because non-impact rotational acceleration from the lateral direction can produce uniform traumatic callosal injury [85,86,87]. The damage seen in corona radiata is also consistent with a rotational shearing mechanism [43].

Unlike many studies, we included AD/RD as a scalar metric, as recommended by others [59] who found that AD/RD in more cylinder-like tracts best differentiated older and younger adults. The addition of AD/RD to the SVM model boosted classification accuracy of bmTBI from 75% to almost 90%. As water diffusivity in corpus callosum and corticospinal tracts are better approximated as cylindrical diffusion, AD/RD is an important metric that should be used in DTI research.

Our DTI feature selection method is biologically reasonable and computationally efficient. The choice of features was based on well-accepted anatomical templates, and our SVM approach focused on regional group differences in white-matter tracts, rather than whole-brain white matter, because injuries in bmTBI are typically focal [88]. By using the average metric of voxels within a tract as a feature to reduce the dimensionality of the data and minimize overfitting of the SVM model [89], this also reduced computational costs, which otherwise become large and difficult to handle [90].

### 4.3. Limitations

Limitations include the small bmTBI and control cohorts, which could bias the selection of abnormal clusters for the SVM analyses. Specifically, we applied feature selection to the whole dataset in a supervised mode, which can bias the SVM approach. Ideally, feature selection is applied using the SVM training dataset, and then SVM performance is evaluated using the testing dataset. The consistency of the extracted DTI features across the cross-validation folds can then be evaluated. This approach was not used in the present study owing to our small sample size, for which instability of the selected features may exist if not using the whole data set. An alternative approach is to test the robustness of the selected features using new samples, which is what we recommend for future studies. In this regard, larger samples may uncover regions of interest not identified in our study. At the same time, our results may be more specific to mild blast injuries owing to a more homogeneous bmTBI cohort than past studies, with only two patients having PTSD and none with depression or other psychiatric conditions. Likewise, age and years education did not differ between our bmTBI and controls, who were also Veterans/military personnel with a history of blast exposure, which adjusts for potential effects of occupational exposure to catastrophic experiences on neurotrauma. Second, owing the high comorbidity of PTSD in veterans with blast injuries, two participants with comorbid bmTBI-PTSD were included in our study, as considerable overlap would be expected in the white-matter tracts that are affected by a blast force, regardless of PTSD. Indeed, a bmTBI study of people with and without comorbid PTSD reported distributed white-matter pathology in tracts that overlapped with our study, yet pathology was not related to a diagnosis of PTSD or depressive symptoms [77]. Moreover, in univariate tract-based analyses we failed to find group differences in FA of the cingulum, which is common in PTSD [34,91,92,93]. Altogether, it is unlikely that our results are confounded by the inclusion of two comorbid participants. Still, relationships between bmTBI and PTSD are not well understood [94], and future studies are needed that compare bmTBI, PTSD, and comorbid cohorts to fully understand similarities and differences in the pathogenesis of white-matter disturbances amongst these conditions. Third, similar to many studies of mTBI or bmTBI [29,39,48,49] nonindependent samples were used to define DTI abnormalities and estimate classification accuracy, which can overestimate diagnostic utility. Our focus on selecting clinically relevant features for modeling may have minimized this problem, together with cross-validation of SVM models wherein consensus features defined diagnostic accuracy [49,51]. Nonetheless, this is the first application of SVM using DTI metrics, thus replication cohorts will be needed to validate the sensitivity and specificity of the optimal SVM model generated in our study. More generally, replication is always important as bmTBI cohorts often differ in the extent to which they present with comorbidities (e.g., PTSD, depression) or other features (e.g., prior head injury, positive MRI) that can affect DTI results. 

Heterogeneous injuries across individuals from diverse mechanical forces may also play a role in discrepant findings. For this reason, some studies employ techniques that quantify normal and abnormal FA throughout spatially heterogeneous white matter [48]. In contrast to our approach, however, classification accuracy is modest (≈75%) possibly due to using FA as the sole metric and/or absence of information about the spatial location of injuries. Our results indicated that the combination of multiple DTI metrics in well-defined tracts is essential for improving diagnostic accuracy. Moreover, the ROIs uncovered in our study rested in frontal tracks, which are the most vulnerable to blast exposures [81,83,85,86,87], and therefore may be the most potent markers of bmTBI, irrespective of abnormalities in other white-matter tract injuries. Future studies comparing tract-based and spatially heterogeneous approaches, which employ multiple DTI metrics, would elucidate the relative value of these approaches in handling diagnostic challenges associated with heterogeneous injuries across individuals.

## 5. Conclusions

DTI biomarkers derived from modeling in the present study SVM differentiated bmTBI and HCs with high sensitivity and specificity. These preliminary findings indicate that the most potent discriminating features rested in anterior tracks for which AD/RD and FA were abnormally increased. The high classification accuracy of these features points to their potential for diagnosis, which we plan to validate in larger, independent cohorts. If our findings are successfully validated, this knowledge could promote treatment interventions such as transcranial magnetic or electrical stimulation. For example, we found that transcranial electrical stimulation treatments in people with chronic mTBI from blast and non-blast injuries markedly reduced or eliminated persistent post concussive symptoms and reduced abnormal slow-waves in approximately the same brain areas that showed abnormal slow-wave generation prior to treatment [95]. Transcranial magnetic stimulation also reduces post-concussive depression, headaches, and may improve cognition [96,97,98,99]. Both stimulation approaches, however, are not typically guided by knowledge about specific regions of brain damage. Rather, stimulation is usually administered at the same anatomical site in all participants, which may limit treatment effectiveness for some patients. Knowledge about the most prominent loci of abnormal white matter disturbances may therefore guide the development of stimulation targets that are optimal for individual patients. Lastly, longitudinal changes in regional diffusion metrics should also be tracked to elucidate mechanisms of recovery, which could also have important implications for clinical interventions.

## Figures and Tables

**Figure 1 diagnostics-12-00987-f001:**
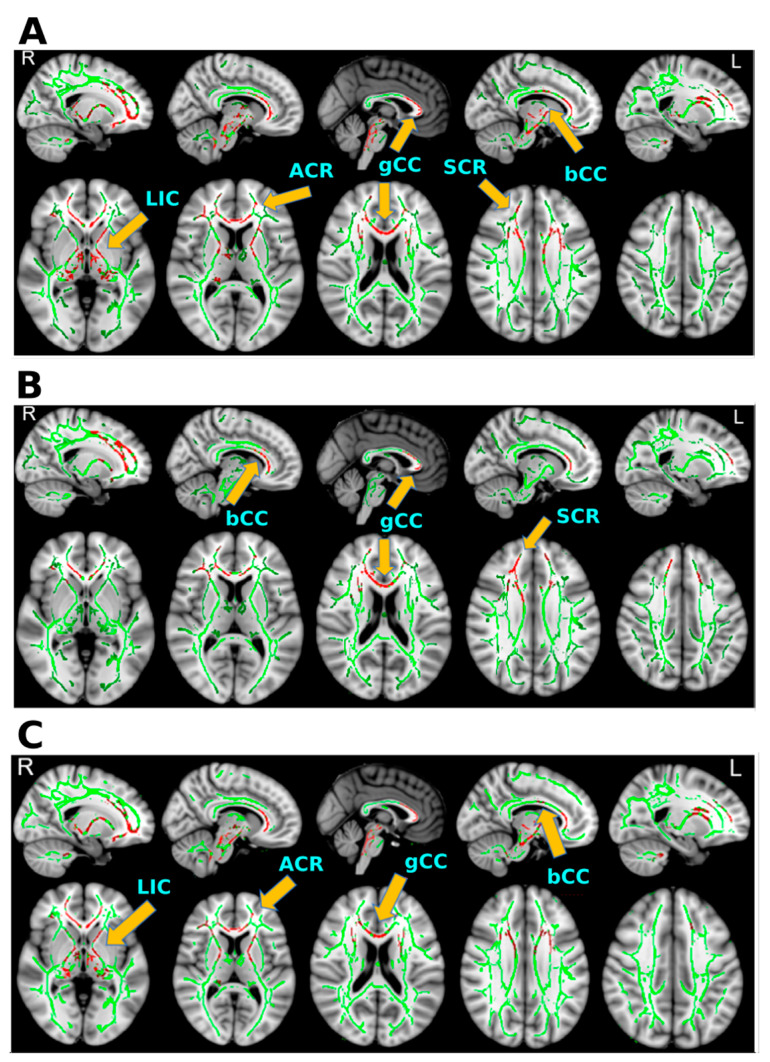
Significant group differences in FA, RD, and AD/RD in white-matter fiber tracts. In all figures, green skeletons show the averaged skeletonized FA of the HC and bmTBI participants on both sagittal (top) and axial (bottom) views. Red regions and arrows/tract labels designate the locations of the voxels that showed statistically significant group differences in a diffusion metric (FEW corrected *p* value < 0.05). (**A**) Tracts showing significantly higher FA in the bmTBI group than the healthy control group. (**B**) Tracts showing significantly decreased RD in the bmTBI group in comparison to the HC group. (**C**) Tracts showing significantly higher AD/RD in the bmTBI group than in the HC group. ACR = anterior corona radiata; bCC = body of corpus callosum; gCC = genu of corpus callosum; LIC = anterior limb of internal capsule; SCR = superior corona radiata.

**Figure 2 diagnostics-12-00987-f002:**
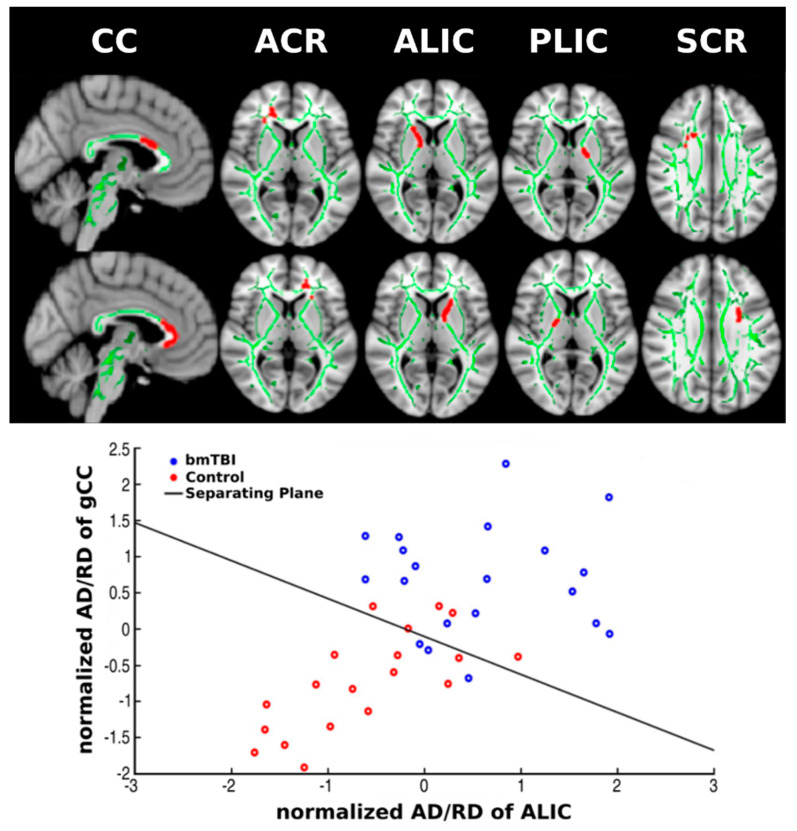
DTI features and inputs into the supervised vector machine (SVM) model. The top figure displays the locations of candidate features that were used to build the stepwise SVM model. Green skeletons are the averaged skeletonized FA of the control and bmTBI groups. Red regions show the mapping of tract locations for the features. The leftmost column (sagittal views) shows the two features generated from the corpus callosum, namely the bCC (top) and gCC (bottom). The remaining columns (axial views) display the anatomical locations of the ACR, ALIC, PLIC, and SCR features for the right (top row) and left (bottom row) hemispheres. The graph below visualizes how a plane separates controls and bmTBI participants in SVM with the *x*-axis being normalized AD/RD of left ALIC and the *y*-axis being normalized AD/RD of gCC. ACR = anterior corona radiata; ALIC = anterior limb of internal capsule; bCC = body of corpus callosum; gCC = genu of corpus callosum; PLIC = posterior limb of internal capsule; SCR = superior corona radiata.

**Table 1 diagnostics-12-00987-t001:** Demographic characteristics and neuropsychological test performances in the control and bmTBI groups.

	Control Group		bmTBI Group			
	Mean	SD	Mean	SD	*p*-Value	Cohen’s *d*
Age	28.00	3.500	27.40	6.227	0.6453	0.1234
Years of education	12.58	0.769	13.05	2.059	0.2635	−0.3324
Months post-injury			10.38	6.271		
D-KEFS						
Number-Letter Sequencing	11.37	1.383	10.55	1.669	0.1049	0.5374
Letter Fluency	10.79	3.326	8.750	3.093	0.0547 ^	0.6356
Category Fluency	11.58	3.097	10.40	2.963	0.2321	0.3894
Category Switching (total correct)	11.79	2.974	9.800	3.002	0.0447 *	0.6660
Category Switching (total switching accuracy)	12.05	2.368	10.75	2.100	0.0938	0.5819
WAIS-III						
Symbol Search	10.79	3.794	10.55	2.417	0.8145	0.0773
Digit Symbol Coding	10.16	2.754	8.800	2.608	0.1222	0.5073

Group differences were tested using independent *t*-tests. Neuropsychological measures are expressed as scaled scores. SD = standard deviation; D-KEFS = Delis-Kaplan Executive Function System; WAIS-III = Wechsler Adult Intelligence Scale Version 3; * *p* < 0.05; ^ subthreshold trend for group differences (0.05 ≤ *p* < 0.1).

**Table 2 diagnostics-12-00987-t002:** Percentage symptoms endorsed on the HISC **^±^** in the bmTBI and control groups.

Symptoms	bmTBI (%)	Control (%)	Symptoms	bmTBI (%)	Control (%)
Headaches	90.0	5.26	Lack of spontaneity	0.00	0.00
Dizziness	70.0	5.26	Affective lability	10.0	5.26
Fatigue	50.0	10.5	Depression	20.0	5.26
Memory difficulty	85.0	15.8	Concentration	10.0	10.5
Irritability	60.0	15.8	Bothered by noise	0.00	0.00
Anxiety	55.0	0.00	Bothered by light	0.00	5.26
Sleep problems	60.0	5.26	Coordination/balance	15.0	10.5
Hearing difficulties	60.0	10.5	Motor difficulty	10.0	0.00
Visual difficulties	10.0	0.00	Speech difficulty	0.00	5.26
Personality changes	25.0	5.26	Numbness/tingling	20.0	0.00
Apathy	5.00	0.00			

^±^ HISC = modified Head Injury Symptom Checklist (Alvin Jr et al., 1984).

**Table 3 diagnostics-12-00987-t003:** Group differences in DTI metrics of white-matter features.

Table	Group	FA	*p* Value	RD ^±^	*p* Value	AD/RD	*p* Value	Voxels
lACR	HC	0.44 ± 0.024	<0.001	0.54 ± 0.027	0.0098	2.04 ± 0.11	<0.001	
	bmTBI	0.47 ± 0.018		0.52 ± 0.021		2.15 ± 0.08		675
rACR	HC	0.45 ± 0.029	<0.001	0.56 ± 0.033	<0.001	2.11 ± 0.13	<0.001	
	bmTBI	0.48 ± 0.024		0.53 ± 0.025		2.24 ± 0.13		1073
lALIC	HC	0.54 ± 0.015	<0.001	0.47 ± 0.018	<0.001	2.63 ± 0.10	<0.001	580
	bmTBI	0.56 ± 0.013		0.45 ± 0.014		2.83 ± 0.10		
rALIC	HC	0.57 ± 0.020	<0.001	0.45 ± 0.023	0.0026	2.80 ± 0.14	<0.001	364
	bmTBI	0.60 ± 0.019		0.43 ± 0.018		3.03 ± 0.16		
lPLIC	HC	0.62 ± 0.017	<0.001	0.41 ± 0.016	0.0608	3.18 ± 0.14	<0.001	
	bmTBI	0.64 ± 0.019		0.40 ± 0.021		3.40 ± 0.23		1107
rPLIC	HC	0.63 ± 0.017	0.0015	0.40 ± 0.017	0.0056	3.29 ± 0.17	<0.001	
	bmTBI	0.65 ± 0.020		0.38 ± 0.020		3.54 ± 0.26		1472
lSCR	HC	0.45 ± 0.027	<0.001	0.53 ± 0.023	0.0058	2.10 ± 0.13	0.0088	
	bmTBI	0.48 ± 0.027		0.50 ± 0.022		2.23 ± 0.15		260
rSCR	HC	0.45 ± 0.022	0.0026	0.53 ± 0.022	0.0215	2.15 ± 0.10	0.0067	
	bmTBI	0.48 ± 0.026		0.51 ± 0.026		2.27 ± 0.14		247
gCC	HC	0.64 ± 0.032	0.0014	0.46 ± 0.043	0.001	3.45 ± 0.32	<0.001	
	bmTBI	0.68 ± 0.024		0.42 ± 0.029		3.94 ± 0.35		222
bCC	HC	0.66 ± 0.028	<0.001	0.43 ± 0.042	<0.001	4.08 ± 0.39	0.0031	
	bmTBI	0.69 ± 0.020		0.39 ± 0.028		4.41 ± 0.32		170

Group differences in diffusion metrics were tested using independent *t*-tests. l = left; r = right; ACR = anterior corona radiata; ALIC = anterior limb of internal capsule; bCC = body of corpus callosum; gCC = genu of corpus callosum; PLIC = poster limb of internal capsule; SCR = superior corona radiata; ^±^ RD is expressed in the unit of mm^2^/s. FA and AD/RD are unitless.

**Table 4 diagnostics-12-00987-t004:** Features and performances of the best SVM model.

Model	Accuracy	Sensitivity	Specificity	Kernel	Features: Weightings (Mean ± Standard Deviation)
FA and AD/RD	89%	90%	88%	Linear	Normalized AD/RD lALIC: (1.43 ± 0.16) AD/RD gCC: (0.84 ± 0.14) AD/RD rALIC: (0.77 ± 0.14) FA rACR: (−1.08 ± 0.15) FA lSCR: (0.69 ± 0.12)

l = left hemisphere; r = right hemisphere; ACR = anterior corona radiata; ALIC = anterior limb of internal capsule; gCC = genu of corpus callosum; SCR = superior corona radiata.

## Data Availability

The data that support the findings of this study are available on request from the corresponding author. The data are not publicly available due to privacy or ethical restrictions imposed by U.S. Department of Veterans Affairs and U.S. Department of Defense.

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
