# Peer review of "Detection of Chronic Blast-Related Mild Traumatic Brain Injury with Diffusion Tensor Imaging and Support Vector Machines"

_diagnostics, 2022, doi:10.3390/diagnostics12040987_

Round 1

Reviewer 1 Report

Why did the authors choose a different number of subjects between TBI and HC? What was the statistical power achieved?

How did the authors decide “negativity” of MRI findings? Which sequences? Which plane? How much thick slices? Confirmation by board-certified neuroradiologists is recommended.

Please include Units wherever appropriate in the Tables.

What is the DTI scan coverage? The authors took DTI scans twice. Was there any scanning parameter set up (e.g. encoding direction) different between the two? What is the acquisition time? Longer scans might result in patent movement. How did the authors define motion artifacts?

Did the sensitivity improve with AD/RD when compared to FA or MD alone? Similarly, did SVM improve the sensitivity which is the authors’ hypothesis?

Were there any voxels that were correlated with the DTI findings?

Are these findings selective of TBI? There have been papers that suggest suicidal attempts when lesions exist in ALIC.

What is the clinical significance of this study?

Provide examples of conventional MRI

Author Response

1. Why did the authors choose a different number of subjects between TBI and HC? What was the statistical power achieved?

Reply: The results from the power analyses were added to the revised manuscript. Power analyses were performed for all the white-matter features listed in Table 3 using G*Power (Faul et al., 2009). All features showed power > 0.80 at two-tailed α=0.05 level, with the exceptions of the lACR’s RD feature (power = 0.71) and lPLIC’s RD feature (power = 0.37).

2. How did the authors decide “negativity” of MRI findings? Which sequences? Which plane? How much thick slices? Confirmation by board-certified neuroradiologists is recommended.

Reply: First, the revised version fully describes the T1, T2, and FLAIR protocols, which were used to ascertain if there where clinically significant findings on MRI scans. Please note that the original version of the manuscript (first paragraph, section 2.1) stated that “MRI scans were reviewed by a board-certified neurologist (RRL) and were negative in all bmTBI participants.” RRL refers to Dr. Roland R. Lee, who is also a co-author on this manuscript.

3. Please include Units wherever appropriate in the Tables.

Reply: Thanks for pointing this out. First, in Table 3 there was a typo in the reporting the unit for RD which should be mm2/sec. FA and AD/RD are unit free. All other tables report the units for data. 

4. What is the DTI scan coverage? The authors took DTI scans twice. Was there any scanning parameter set up (e.g. encoding direction) different between the two? What is the acquisition time? Longer scans might result in patent movement. How did the authors define motion artifacts?

Reply: Thanks for this question. We added the requested information to sections 2.2 and 2.3. The DTI scan provided whole brain coverage (56 slices, 2.5 mm thickness). The same number of slices with no gap were used across all subjects. Data were acquired axially starting just above the brain. Since we did not employ oblique scanning, the encoding directions were identical across all scans for each subject. The acquisition time for each scan was 13 minutes. 

For each subject, images from each DTI acquisition were concatenated into a single dataset and corrected for eddy currents and motion by registering all brain volumes to the first non-diffusion weighted image using a rigid body transformation (no rotation). By avoiding rotations in our motion correction (i.e., we only considered motion in the X,Y, and Z direction), we also avoid issues with changing the encoding direction prior to analysis.

Each DTI scan was visually inspected to ensure the absence of susceptibility artifacts and significant rotation in the brain volume. If significant motion was present, the entire brain volume was removed from the dataset. The revised Supplementary Materials 2.0 provides additional details about motion correction procedures.

5. Did the sensitivity improve with AD/RD when compared to FA or MD alone? Similarly, did SVM improve the sensitivity which is the authors’ hypothesis? The core concept of the procedure is to try all possible combinations of the DTI metric set and select the optimal combination(s).

Reply: The revised results section was clarified as it pertains to the features that were entered into the SVM models.  First, MD and AD were not included in the SVM model, since differences between the bmTBI and HC groups were not found for these metrics across any of the white- matter tracts (see Figure 1 and Table 3). As for the SVM models, FA features alone showed 75% accuracy, 82% sensitivity, and 79% specificity. By including AD/RD into the model, 89% accuracy, 90% sensitivity, and 88% specificity were obtained. As for the reviewer’s question about whether SVM improved the sensitivity, the SVM model itself demonstrates that multiple features were needed to optimize sensitivity. In other words, no single feature was sufficient for obtaining the same level of sensitivity.

6. Were there any voxels that were correlated with the DTI findings?

Reply:  The revised SVM methods (section 2.4) states that the mean DTI metric was computed for each cluster that showed significant group differences, which served as the inputs into the SVM models.  In other words, SVM was not conducted on voxels within white-matter clusters.

7. Are these findings selective of TBI? There have been papers that suggest suicidal attempts when lesions exist in ALIC.

Reply: This is an interesting question given the overlap in symptomatology amongst several mental health conditions. While we excluded subjects with current suicide ideation, it is possible that changes in specific suicidal attempts or other clinical conditions (see reply #1 to Review 2) could produce abnormal white-matter in tracts ALIC, as it did in our bmTBI cohort. However, it is another matter as to whether the optimal SVM features identified in our study would also characterize other clinical conditions. In other words, out results highlight the multivariate combination of features in distinguishing bmTBI.  To our knowledge this has not been investigated using multivariate machine learning approaches in people with suicidal attempts or other common co-morbid clinical conditions.   

8. What is the clinical significance of this study?

Reply: Per the reviewer’s request, we have elaborated upon the potential clinical significance of this study in the Conclusions. 

If successfully validated, this knowledge could promote targeted interventions such as transcranial magnetic or electrical stimulation for treatment of post-concussive symptoms, for which effective alternative treatments are limited.  For example, we found that transcranial electrical stimulation treatments in people with chronic mTBI from blast and non-blast injuries markedly reduced or eliminated persistent post concussive symptoms and reduced abnormal slow-waves in approximately the same brain areas that showed abnormal slow-wave generation prior to treatment (Huang et al., 2017). Transcranial magnetic stimulation also reduces post-concussive depression, headaches, and can improve cognition (Lee & Kim, 2018; Leung et al., 2016; Mollica et al., 2021; Vaninetti et al., 2021). Both stimulation approaches, however, typically are not guided by knowledge about specific regions of brain damage. Rather stimulation is frequently administered at the same anatomical site in all participants, which may limit treatment effectiveness in some patients. Knowledge about the most prominent loci of abnormal white-matter disturbances may therefore guide the development of stimulation targets that are optimal for individual patients. Lastly, longitudinal changes in regional diffusion metrics should also be tracked to elucidate mechanisms of recovery, which could also have important implications for clinical interventions.

9. Provide examples of conventional MRI

Reply: we add the conventional MRI sequences to the Introduction, the details of these sequences were addressed in reply to comment 1, Reviewer 1.

Author Response

The authors present their manuscript entitled “Detection of Chronic Blast-related Mild Traumatic Brain Injury with Diffusion Tensor Imaging and Support Vector Machines” for possible publication in Diagnostics. This study enrolled 20 bmTBI and 19 healthy controls who were male combat-deployed active-duty service members or veterans. DTI data were analyzed using TBSS, and the results showed that patients with bmTBI had increased FA, decreased RD, and higher AD/RD indices than the HC group. The extracted FA, RD, AD/RD features were further utilized to discriminate bmTBI from HC subjects using SVM approach, and the performance was up to 89% with a model of FA and AD/RD. In general, the manuscript is well prepared, but there are some issues that need to be addressed or re-analyzed.

Major issues:

1) Line 118. Two patients had a comorbid of PTSD. It was demonstrated that patients with PTSD exhibited decreased FA than those without PTSD [1], so the comorbid of PTSD in bmTBI patients may have influenced the statistical results. I suggest authors to exclude the two patients with PTSD in the statistics.

[1] Fani, N., King, T., Jovanovic, T. et al. White Matter Integrity in Highly Traumatized Adults With and Without Post-Traumatic Stress Disorder. Neuropsychopharmacol 37, 2740–2746 (2012). https://doi.org/10.1038/npp.2012.146

Reply: We excluded bmTBI patients with many mental health comorbidities that can influence DTI outcomes (e.g., depression, suicidal ideation, substance use disorder), owing to the inconclusive findings across studies. However, in the Veteran population there is a high comorbidity of PTSD the bmTBI. Therefore, two bmTBI participants were included in our study as considerable overlap would be expected in the white-matter tracts that are affected by a blast force, regardless of PTSD. As such, including these two patients is appropriate since the primary aim of our study is to distinguish the most prominent DTI features of blast-related injuries. In this regard, a bmTBI study of people with and without comorbid PTDS reported distributed white-matter pathology in tracts that overlapped with our study but also other tracts, yet white-matter integrity was not related to a diagnosis of PTSD or depressive symptoms (Morey et al., 2013). If there were other features specific to PTSD, we would not expect them to emerge in a study with so few comorbid patients (see next paragraph).  While the onset of PTSD is frequently associated with depression, alcohol/drug use, and suicidal ideation, the two patients with comorbid bmTBI/PTSD in our study did not exhibit these mental health conditions. Altogether, we think that it is unlikely that our findings showing prominent DTI features that distinguish blast injuries are confounded by to comorbid PTSD in two patients. Still, the relationships between mTBI and PTSD is not well understood (Kaplan et al., 2018) and further studies are needed. In the revised discussion (4.3 Limitations), this matter is now addressed.

That said, the reviewer commented that FA is decreased in PTSD, presumably suggesting that the two comorbid patients in our study would show the opposite pattern of FA despite incurring a blast injury. Looking to studies of PTSD without TBI, inconclusive findings have been reported regarding whether FA is decreased or increased (Abe et al., 2006; Bazarian et al., 2013; Fani et al., 2012; Weis, Belleau, Pedersen, Miskovich, & Larson, 2018), and while aberrant FA in the cingulum is often reported (Davenport, Lim, & Sponheim, 2015; Fani et al., 2012; O'Doherty et al., 2018; Weis et al., 2018), null results have been reported by others (Dennis et al., 2021). Moreover, aberrant white matter is found in other tracts in PTSD, some of which were identified by our optimal SVM model, including the corpus callosum (Dennis et al., 2021; O'Doherty et al., 2018) and corona radiata (Aschbacher et al., 2018). Altogether, results in PTSD are inconclusive. Future studies leveraging multiple DTI metrics are needed that compare bmTBI, PTSD, and comorbid cohorts to fully understand similarities and differences to understand the pathogenesis of white-matter disturbances. 

2) Line 159. Two DTI acquisition was performed for each subject to increase signal-to-noise ratio and ensure at least one artifact-free scan. What do authors do if two scans were corrupted for the same direction DWIs? This must be described in more details.

Reply: The following narrative was added to the Supplementary Materials 2.0.  “The imaging protocol was designed to avoid issues with insufficient data when removing volumes from the concatenated dataset due to the presence of artifacts or motion. Our DTI protocol consisted of acquiring 5 non-diffusion weighted images (b=0) and 51 diffusion directions (b=1000 s/mm2, 56 brain volumes). We inspected our analysis path and found that on average we removed 1-2 volumes from the concatenated DW data for each subject (1-2 volumes out of 112 brain volumes). We had two subjects in which we had to remove 5 volumes. For these subjects, these volumes were either in the first or the second acquisition. Consequently, we had data acquired at 51 diffusion encoding directions for each subject and avoided corruption of the same diffusion encoding direction within a scan.

3) Line 208. Authors performed cross validation by leaving five subjects out (k=5). I would like to know what are the numbers of bmTBI and HC in the five subjects? Were five subjects randomly selected from all 39 (19 HC + 20 bmTBI) subjects? Since the group sizes are similar between the two groups, I suggest to leave 6 subjects out with 3 bmTBI and 3 HC in cross validation. The results using the balanced datasets will be more reliable.

Reply: Our study employed a conventional, well-accepted cross-validation approach that is specifically suitable for the segmentation of validation data in small datasets. The revision clarifies that the cross-validation procedure randomly selects five samples out of the combined bmTBI and HC cohorts. This cross-validation procedure is performed 10,000 times to calculate the averaged feature weights, correctness, sensitivity, and specificity across validations. We respectfully disagree that this conventional approach is biased (i.e., random sampling, conducted 10,000 times). This is clearly a robust cross validation procedure, so there is no need to adopt another method, especially one that is unconventional and untested. 

4) Line 399. Conclusion needs to be re-written. Authors should conclude their study based solely on their findings.

Reply: This section has been re-written. In addition, the revision further emphasizes that the DTI biomarkers derived from this study are based on the results from the present study, which need to be validated.

Minor issues:

1) Line 126. VA/VOD should be spelled out.

Reply: Change made.

2) Lines 182 and 216. FEW should be corrected to FWE.

Reply: Changes made.

3) Line 184. MNI15 should be corrected to MNI152.

Reply: Change made.

4) Line 226. I don’t see any magnified image in this figure.

Reply: Thanks for catching this.  The revised sentence omits this statement.

5) Line 235. Statistical tool that used in this study should be described.

Reply: The statistical software used for correlation analyses is now specified (SPSS 28).

6) Line 281. Please remove “.”

Reply: The period on line 282 is correct. However, the following sentence was modified to increase the clarity of the results in Table 4.

References:

Abe, O., Yamasue, H., Kasai, K., Yamada, H., Aoki, S., Iwanami, A., . . . Ohtomo, K. (2006). Voxel-based diffusion tensor analysis reveals aberrant anterior cingulum integrity in posttraumatic stress disorder due to terrorism. Psychiatry Res, 146(3), 231-242. doi:10.1016/j.pscychresns.2006.01.004

Aschbacher, K., Mellon, S. H., Wolkowitz, O. M., Henn-Haase, C., Yehuda, R., Flory, J. D., . . . Mueller, S. G. (2018). Posttraumatic stress disorder, symptoms, and white matter abnormalities among combat-exposed veterans. Brain Imaging Behav, 12(4), 989-999. doi:10.1007/s11682-017-9759-y

Bazarian, J. J., Donnelly, K., Peterson, D. R., Warner, G. C., Zhu, T., & Zhong, J. (2013). The relation between posttraumatic stress disorder and mild traumatic brain injury acquired during Operations Enduring Freedom and Iraqi Freedom. J Head Trauma Rehabil, 28(1), 1-12. doi:10.1097/HTR.0b013e318256d3d3

Davenport, N. D., Lim, K. O., & Sponheim, S. R. (2015). White matter abnormalities associated with military PTSD in the context of blast TBI. Human Brain Mapping, 36(3), 1053-1064. Retrieved from https://onlinelibrary.wiley.com/doi/epdf/10.1002/hbm.22685

Dennis, E. L., Disner, S. G., Fani, N., Salminen, L. E., Logue, M., Clarke, E. K., . . . Morey, R. A. (2021). Altered white matter microstructural organization in posttraumatic stress disorder across 3047 adults: results from the PGC-ENIGMA PTSD consortium. Mol Psychiatry, 26(8), 4315-4330. doi:10.1038/s41380-019-0631-x

Fani, N., King, T. Z., Jovanovic, T., Glover, E. M., Bradley, B., Choi, K., . . . Ressler, K. J. (2012). White matter integrity in highly traumatized adults with and without post-traumatic stress disorder. Neuropsychopharmacology, 37(12), 2740-2746. doi:10.1038/npp.2012.146

Faul, F., Erdfelder, E., Buchner, A., and Lang, A.-G. (2009). Statistical power analyses using G*Power 3.1: tests for correlation and regression analyses. Behav. Res. Methods 41, 1149–1160.

Kaplan, G. B., Leite-Morris, K. A., Wang, L., Rumbika, K. K., Heinrichs, S. C., Zeng, X., . . . Teng, Y. D. (2018). Pathophysiological Bases of Comorbidity: Traumatic Brain Injury and Post-Traumatic Stress Disorder. J Neurotrauma, 35(2), 210-225. doi:10.1089/neu.2016.4953

Morey, R. A., Haswell, C. C., Selgrade, E. S., Massoglia, D., Liu, C., Weiner, J., . . . McCarthy, G. (2013). Effects of chronic mild traumatic brain injury on white matter integrity in Iraq and Afghanistan war veterans. Human Brain Mapping, 34(11), 2986-2999. Retrieved from https://onlinelibrary.wiley.com/doi/epdf/10.1002/hbm.22117

O'Doherty, D. C. M., Ryder, W., Paquola, C., Tickell, A., Chan, C., Hermens, D. F., . . . Lagopoulos, J. (2018). White matter integrity alterations in post-traumatic stress disorder. Hum Brain Mapp, 39(3), 1327-1338. doi:10.1002/hbm.23920

Weis, C. N., Belleau, E. L., Pedersen, W. S., Miskovich, T. A., & Larson, C. L. (2018). Structural Connectivity of the Posterior Cingulum Is Related to Reexperiencing Symptoms in Posttraumatic Stress Disorder. Chronic Stress (Thousand Oaks), 2. doi:10.1177/2470547018807134

Reviewer 3 Report

The authors reported very interesting DTI-based findings related to blast-related mild traumatic brain injury.

1) The main limitation of the study is the small number of subjects and it was stated clearly by the authors. In contrast, it is not easy to collect such participants for a clinical study.

2) The main issue with the study is related to feature selection and the SVM machine learning approach.

A. The authors applied a statistical test over the white-matter tracts in order to find clusters of voxels that showed group differences.

My major questions are the following:

1) How did you summarize the DTI cluster-based metrics as a feature set?

   Did you get the mean within every cluster?

2) The selection of features outside the cross-validation approach is a strong bias. You applied feature selection on the whole dataset in a supervised mode.

If you want to report classification accuracies based on DTI metrics, you

have to apply the cluster-based approach in the training set, and then predicted with the extracted features the label of the test set.

Finally, you can show the consistency of the extracted DTI metric-based tracts across the folds.

3) The total number of features is only reported in the discussion part. You should report in the results section.

(28 DTI features)

Author Response

The authors reported very interesting DTI-based findings related to blast-related mild traumatic brain injury.

1) The main limitation of the study is the small number of subjects and it was stated clearly by the authors. In contrast, it is not easy to collect such participants for a clinical study.

Reply. Thank you for recognizing the challenges associated with recruiting a relatively homogeneous bmTBI sample of Veterans.

2) The main issue with the study is related to feature selection and the SVM machine learning approach.

  1. The authors applied a statistical test over the white-matter tracts in order to find clusters of voxels that showed group differences.

My major questions are the following:

1) How did you summarize the DTI cluster-based metrics as a feature set?

   Did you get the mean within every cluster? 

Reply: Voxel-wise statistical analyses first identified group differences in thresholded clusters for each DTI metric. The revision (section 2.4, 1st paragraph) states that the mean DTI metric (i.e., averaged across voxels within the cluster mask) was computed for each cluster that showed significant group differences, which served as the inputs into the SVM models.

2) The selection of features outside the cross-validation approach is a strong bias. You applied feature selection on the whole dataset in a supervised mode. If you want to report classification accuracies based on DTI metrics, you have to apply the cluster-based approach in the training set, and then predicted with the extracted features the label of the test set.

Finally, you can show the consistency of the extracted DTI metric-based tracts across the folds.

Reply: We agree with the reviewer’s assessment. The revised Limitations section (4.3) discusses this issue. The main reason we did not adopt the approach suggested by the reviewer suggested is related to our small sample size, in which instability of the selected features may exist if not using the whole data set. Alternatively, the robustness of the selected features couple be evaluated using new samples, which is what we suggest for future studies.

3) The total number of features is only reported in the discussion part. You should report in the results section. (28 DTI features)

Reply: We apologize for the typo in Table 3. There were 29 total features. This error has been corrected in the revision (i.e., Abstract, Results 3.3, Discussion 4.2).   

Round 2

Reviewer 1 Report

RT 2. Why didn’t the authors include T2*WI to exclude minute hemorrhage? Please include sample images so that a reader can confirm the negativity. A neurologist and a neuroradiologist are different. 

Author Response

Comment: Why didn’t the authors include T2*WI to exclude minute hemorrhage? Please include sample images so that a reader can confirm the negativity. A neurologist and a neuroradiologist are different.

Reply. First, we apologize for the typo pertaining to Dr. Roland Lee (page 6). He is a board-certified neuroradiologist who reviewed the MRI scans. 

Second, the revision reports the T2* weighted protocol, which is sensitive to hemorrhages, which was obtained for all the subjects in the study; and its MRI parameters.  This conventional sequence is also mentioned in the Introduction. Sample images (T1, T2, T2*, and FLAIR) from two patients are displayed in Supplementary Figure 1.

As mentioned in the text, the MRI images were negative for all subjects; including the T2* images which were negative for intracranial hemorrhage in all subjects.

Reviewer 2 Report

The manuscript is improved after revision. However, some abbreviations need to be correctly defined, such as incorrect "FEW", FMRIB, ICBM, etc.

Author Response

Comment: The manuscript is improved after revision. However, some abbreviations need to be correctly defined, such as incorrect "FEW", FMRIB, ICBM, etc.

Reply. Thank you!  Typo’s have been corrected throughout the manuscript.  Please note that FSL is the correct abbreviation for the Functional Magnetic Resonance Imaging of the Brain Software Library, which is spelled out in the revision.  All FEW have been changed to FEW which stands for familywise error rate (first defined on page 9). FLAIR is spelled out in the introduction. Other abbreviations are also spelled out (e.g., ICBM, MNI, SPSS).

Reviewer 3 Report

The authors answered properly to my suggestions.

I have no further comments.

I recommend the acceptance of the manuscript in its present form.

Author Response

Thank you!